# D-Pinitol—Active Natural Product from Carob with Notable Insulin Regulation

**DOI:** 10.3390/nu14071453

**Published:** 2022-03-30

**Authors:** Abdullatif Azab

**Affiliations:** Carobway Ltd., Nes Ziona 7406520, Israel; abedazab@carobway.com; Tel.: +972-50-5650025

**Keywords:** carob, inositols, D-Pinitol, medicinal activities, antidiabetic, insulin regulator, mechanism of action

## Abstract

Carob is one of the major food trees for peoples of the Mediterranean basin, but it has also been traditionally used for medicinal purposes. Carob contains many nutrients and active natural products, and D-Pinitol is clearly one of the most important of these. D-Pinitol has been reported in dozens of scientific publications and its very diverse medicinal properties are still being studied. Presently, more than thirty medicinal activities of D-Pinitol have been reported. Among these, many publications have reported the strong activities of D-Pinitol as a natural antidiabetic and insulin regulator, but also as an active anti-Alzheimer, anticancer, antioxidant, and anti-inflammatory, and is also immune- and hepato-protective. In this review, we will present a brief introduction of the nutritional and medicinal importance of Carob, both traditionally and as found by modern research. In the introduction, we will present Carob’s major active natural products. The structures of inositols will be presented with a brief literature summary of their medicinal activities, with special attention to those inositols in Carob, as well as D-Pinitol’s chemical structure and its medicinal and other properties. D-Pinitol antidiabetic and insulin regulation activities will be extensively presented, including its proposed mechanism of action. Finally, a discussion followed by the conclusions and future vision will summarize this article.

## 1. Introduction

### 1.1. Carob: The Faithful Companion of Humanity

Carob (*Ceratonia siliqua* L.) is one of the most important nutritional crops for peoples of the Middle East, North Africa, and Southern Europe [1]. Carob fruits (named pods or kibbles), contain a wide range of macro- and micronutrients, as well as many other natural products. A summary of Carob fruit composition is presented in Table 1.

However, since antiquity, humans have used different parts of the Carob tree for many and interesting purposes [3]. Among these, analgesic and anti-inflammatory activities are the most important [4,5]. Most of the traditional medicine uses have utilized different forms of fruits, including unripe pods, but these utilizations included extracts, decoctions and infusions of leaves and bark [6].

Modern research followed traditional knowledge and dozens of studies were published to date about dozens of medicinal activities of Carob’s various products, including its extracts and single natural products. Consequently, many review articles that summarize the research articles can also be found [7,8,9,10]. However, one of the best review articles about Carob’s composition has been published by K. Rtibi et al., where they focus on Carob-derived treatments of the gastrointestinal tract [11]. In Figure 1, major and new (red names) phenolic compounds are shown [12,13,14].

To conclude this section, it is important to indicate that in recent years there has emerged a rapidly growing interest in Carob seeds, their composition (protein rich), nutritional potential and medicinal activities [2,15,16].

### 1.2. Insulin Resistance in Type 2 Diabetes

Type 2 diabetes (T2D) is defined by the World Health Organization (WHO) as a “metabolic disorder of multiple etiology characterized by chronic hyperglycemia with disturbance of carbohydrate, fat, and protein metabolism resulting from defects in insulin secretion, insulin action, or both” [17]. The International Diabetes Federation reported that in 2018, there were 463 million people around the world affected by this disease, and the organization estimates that by 2045, there will be 700 million people affected by it [18]. It has also been reported that in 2017, the global healthcare expenditure associated with diabetes and its complications was USD 850 billion. The prevalence rate is estimated as 13.5% in low-income countries, compared with 10.4% in high-income nations. It is interesting to mention the fact that this trend is also found within different ethnicities in the same country. In the USA, the ethnic distribution of T2D follows the “rule” higher-income-lower-diabetes: Non-Hispanic Whites (highest income) 7.6%, Asians 9%, Hispanics 12.8%, and African Americans (lowest income) 13.2%, in 2017 [19]. In Israel, the author’s home country, there are 12% of diabetics among Arabs (lower income) and 6.2% among Jews (higher income) [20].

Therefore, in the abovementioned definition of T2D, insulin plays a critical role, and “insulin resistance” is the major cause of this disease. This health disorder is defined as: “a defect in insulin-mediated control of glucose metabolism in tissues—prominently in muscle, fat and liver” [17], but insulin has various functions in the human body, and they are presented in Table 2 [21].

The mechanism of action of insulin in healthy conditions can be found in many publications [22], and a simplified illustration of it is shown in Figure 2.

Insulin enters the cell through an insulin receptor. As a result, tyrosine (Tyr) phosphorylation occurs on the insulin receptor substrate (IRS) protein. The resulting adduct activates phosphoinositide 3-kinase (PI3K), resulting in activation of phosphoinositide-dependent kinase-1,2 (PDK1/2). Protein kinase (AKT) gets phosphorylated by PDK1/2 and promotes glucose transporter 4 (GLUT4) translocation to plasma membrane and facilitates glucose into cell. Thioredoxin interacting protein (TXNIP) inhibits is blocked.

Numerous research articles have been published about this key factor of T2D, and dozens of review articles that summarize these research publications. However, it is important to understand the possible mechanisms of insulin resistance that were also presented in most of these scientific publications. One of the most comprehensive and illustrated review articles was published by M.C. Petersen and G.I. Shulman [23]. Insulin resistance is discussed as major and sub-major types, where each section is illustrated with many figures and graphs.

The review article of H. Yaribeygi et al. follows the previous reference, though it is far less comprehensive [24]. However, one of its clearest advantages is the table that summarizes the molecular mechanisms that are involved in insulin resistance (page 6 in Ref. [24]), and it is partially cited here as shown in Table 3.

The review article of D.E. James has special importance for two major reasons [26]. First, it includes excellent figures that explain the putative factors that contribute to insulin resistance (Figure 4, page 12 in Ref. [26]). Second, it discusses the situation of fasting in insulin resistance conditions. This situation has great relevance for hundreds of millions of people around the world. Another review article with special importance about insulin resistance has been recently published by W.A. Banks and E.M. Rhea [27]. This article is important for three major reasons. First, it links insulin resistance with the brain–blood barrier (BBB); second, it discusses the relation of insulin signaling and oxidative stress manifestation in T2D and Alzheimer’s disease; and third, it contains excellent illustrations, especially the figure that shows the interactions between insulin and oxidative stress.

### 1.3. Treatment of Insulin Resistance with Natural Products

As mentioned in the previous section, T2D is a severe global health issue and a major cause of financial burden. Consequently, many methods have been developed to target this disorder. However, before presenting treatments that are based on natural products, we will briefly present some selected synthetic pharmaceuticals.

C.L. Reading et al. have reported the anti-inflammatory activity and improvement of the insulin-sensitivity activity of synthetic sterol (Figure 3) in insulin-resistant obese-impaired glucose tolerance [28].

A significantly different approach has been reported by S. Xue et al. who report a treatment for hepatogenous diabetes using Oleanolic acid, which triggered the expression of short-peptide genetic synthesis [30]. The synergistic activity of Oleanolic acid and the peptide (researchers have named it shGLP-1), proved to be more efficient than the activity of each component separately. To conclude this part, we indicate the review article of R. Vieira et al. which is very informative and comprehensive [31].

Many natural products have been tested and published for their insulin regulation activity. F.S. Saadeldeen et al. list in their excellent review article 98 naturally occurring compounds that regulate glucose metabolism and treat insulin resistance [32]. This article provides the structure of each compound, its botanical source, and its activity.

Following traditional Chinese medicine therapeutic methods, J. Li et al. list pure natural products and herbal formulations used to treat insulin resistance [33]. Formulations are listed with their Chinese names, and detailed information about methods and purposes of use.

In addition to D-Pinitol, which will be discussed in Section 3, numerous natural products have been published in research articles for having insulin regulation activity. We limit our presentation here to two of these compounds that have been mentioned in very recent publications. First, R. Alaaeldin et al. reported the amelioration of insulin resistance of Carpachromene (Figure 4), a natural product that can be found in Banyan (*Ficus binghalensis* L.) [34].

They found (in vitro model) that Carpachromene has significant insulin resistance amelioration compared with Metformin, a synthetic drug widely used for treating this disorder.

The second report was published by A. Deenadayalan et al. who tested the effect of Stevioside (Figure 5) on insulin resistance, in both in vivo (rats) and in silico models [35].

Their findings indicate that this compound has similar activity to metformin.

Finally, it is important to mention very recent research published by H. Sanz-Lamora et al. that found that treatment with pure polyphenol supplementation (D18060501) worsened insulin resistance in diet-induced obese mice [36].

## 2. Inositols—A Brief Presentation

Inositols are naturally occurring *Cyclitols* or *Polyols*, and they can be found in mammalian and plant kingdoms [37]. In terms of more specific chemical structure, these natural products are stereoisomers of hexahydroxy cyclohexane. In Figure 6, the structures of naturally occurring inositols are shown.

The biological properties of inositols have been extensively studied and published. Most of these activities have been summarized by O.C. Watkins et al. [38]. These properties include insulin regulation, antidiabetic, antioxidant, antibacterial, female fertility enhancer, metabolic syndrome treatment, antidepressant, gastroprotective, hepatoprotective, hypolipidemic and antiaging. However, in this review and in most published literature about the properties of these compounds, it is clear that most studies have focused on two activities: insulin regulation and treatment of female fertility disorders. In Table 4 we cite some of these notable publications, in chronological order.

From Carob, six inositols and their derivatives (methyl ethers) were isolated and characterized [61]. Their structures are shown in Figure 7.

*myo*-Inositol is the most abundant compound of this family in all life forms, followed by D-Pinitol and its precursor, D-*chiro*-Inositol, in the plant kingdom. D-Bornesitol and D-Sequoyitol are relatively rare, and their properties are almost unknown. D-Ononitol has been very limitedly studied [62,63].

## 3. D-Pinitol: Occurrence, Isolation, and Properties

D-Pinitol can be found in more than 20 plant sources, and its highest content is in Carob pods, at 5.5% [2,64]. To date, more than 40 publications have reported the quantification and/or isolation of this important natural product. One of the most notable works has been published by O. Negishi et al. [65]. They determined the content of methylated inositols in 43 edible plants by the HPAE-PAD analytical method. J. Qiu et al. reported the determination of D-Pinitol in rat plasma [66]. This study is highly important since it provides understanding of the pharmacokinetics and bioavailability of D-Pinitol in vivo.

The medicinal and other properties of D-Pinitol have been extensively studied and published. In Table 5, we list most of these reports, excluding publications that report no or low results.

## 4. D-Pinitol as Insulin Regulator

In Section 3, we cited eight important published studies about the activity of D-Pinitol as insulin regulator (Table 5). In fact, the number of publications about this topic is much higher, and many review articles have published about it and other medicinal properties of D-Pinitol. These review articles and the research publications that they cite, conclude that D-Pinitol has two mechanisms of action as an insulin regulator [152]: insulin sensitizing and insulin mimetic.

K. Srivastava et al. present the insulin-sensitizing effect of D-Pinitol in their review article about this natural product [153], and a simplified illustration of this effect is shown in Figure 8.

Interestingly, in a table that lists the botanical sources of D-Pinitol in Ref. [153] (page 3), the authors do not mention the three plants with the highest content of this natural product: Carob, Bougainvillea and Soybean [64].

T. Antonowski et al. present the insulin-like (insulin-mimetic) activity of D-Pinitol [154]. This publication, and others, demonstrates the simplified mechanism shown in Figure 9.

This minireview article is notably useful for understanding the structures of cyclitols and their role in ameliorating metabolic syndrome and diabetes.

## 5. Discussion

D-Pinitol is a naturally occurring inositol that can be found in many plant species. Carob has the highest content of D-Pinitol, which has a wide range of medicinal and other properties (Section 3). One of these, and probably the most important, is insulin regulation, which has two major mechanisms: insulin-sensitizing and insulin-mimetic [152].

Many natural products have one or both properties of insulin regulation, including plant extracts and other mixed compounds. For example, S.A. Kalekar et al. have reported on the in vitro insulin-sensitizing activity of hydroethanolic extracts of three plants: *Phyllanthus emblica* L., *Tinospora cordifolia* (Thunb.) Miers and *Curcuma longa* L. [155]. In a more recent study, V. Stadlbauer et al. tested more than 600 plant extracts and found three of them to have clear in vivo insulin-mimetic activity: *Xysmalobium undulatum* L., *Sapindus mukorossi* L., *Chelidonium majus* L. [156]. It is important to mention that in this study Carob is not included.

Despite the abovementioned, D-Pinitol, and D-Pinitol-containing products of Carob, have several advantages over other insulin-regulating plant products, due to the following reasons:(A)D-Pinitol content of Carob (pods) is the highest of all plants [64].(B)D-Pinitol-containing products of Carob such as molasses, have important health benefits [157].(C)Compared with most other natural products that have insulin-regulation activity, such as polyphenols, D-Pinitol is more stable in biological gastric conditions [48]. This property increases its bioavailability in the human body.(D)In addition to that which is mentioned in C, D-Pinitol is generally stable, but even if it undergoes methoxy group hydrolysis, the resulting compound is *chiro*-Inositol, which is an active insulin-regulator as well [158]. See Figure 6.(E)Even though there is a limited number of studies that indicate it, it is evident that D-Pinitol’s activities are significantly increased when it synergistically acts with other natural products [25,92,150,159].(F)D-Pinitol has wide range of medicinal activities (Table 5), so it is a multi-functional natural product. This property increases its potential as a drug.

## 6. Conclusions and Future Horizons

Most of the medicinal properties of D-Pinitol have been studied and published. Some of these have been extensively investigated, while others were limitedly or even not published. It is very important to conduct further studies of all activities of D-Pinitol, but activities such as insulin regulation, anti-Alzheimer, antiaging and possible anti-Parkinson activities must draw more attention.

The synergistic effect of D-Pinitol with other natural products of Carob and other plants is in its beginnings, so this topic must also be thoroughly studied.

Our group is currently investigating some known and unpublished activities of D-Pinitol, and we are examining possible clinical and other applications that will hopefully lead to healthy foods, food-additives, and other healthy products.

## Figures and Tables

**Figure 1 nutrients-14-01453-f001:**
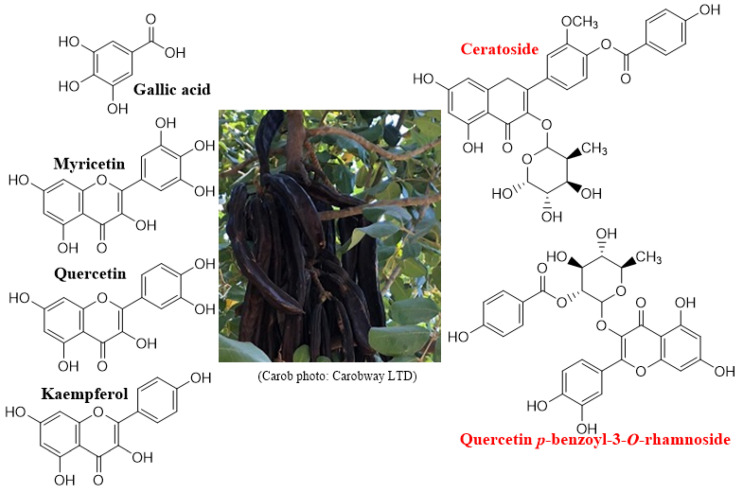
Major and new phenolic compounds found in Carob pods and leaves [12,13,14].

**Figure 2 nutrients-14-01453-f002:**
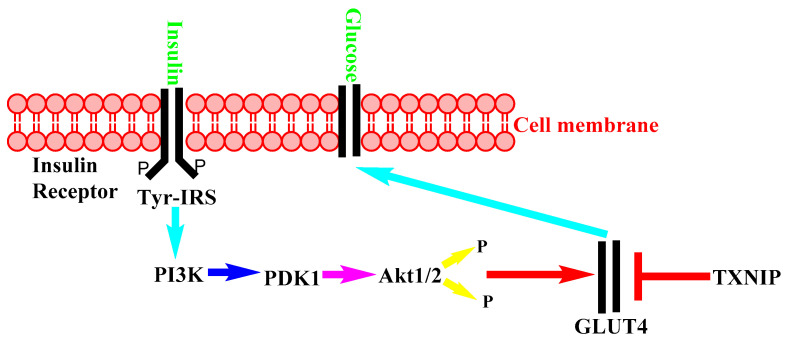
Insulin mechanism of action in healthy conditions.

**Figure 3 nutrients-14-01453-f003:**
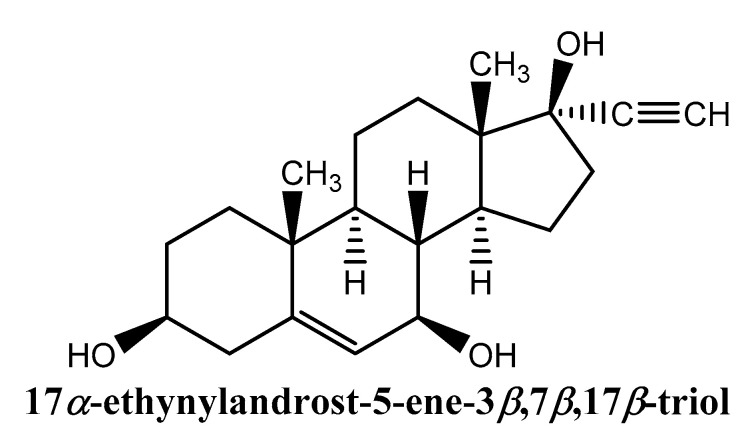
Synthetic sterol with insulin-sensitivity improvement activity [29].

**Figure 4 nutrients-14-01453-f004:**
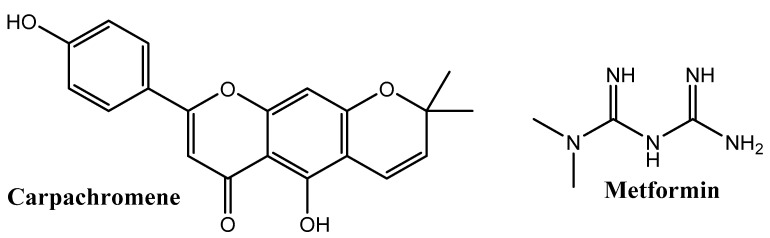
Carpachromene and Metformin.

**Figure 5 nutrients-14-01453-f005:**
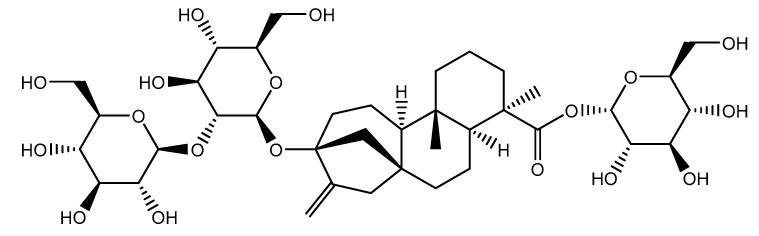
Stevioside (*Stevia rebaudiana* Bertoni).

**Figure 6 nutrients-14-01453-f006:**
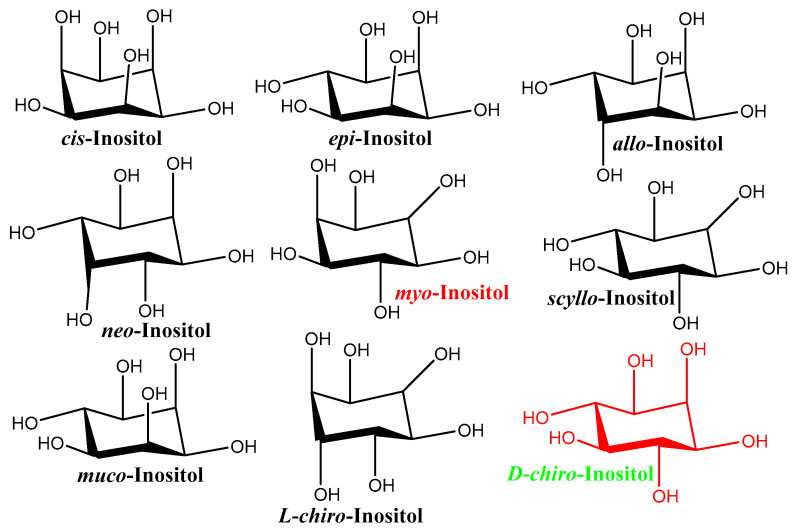
Structures of naturally occurring inositols.

**Figure 7 nutrients-14-01453-f007:**
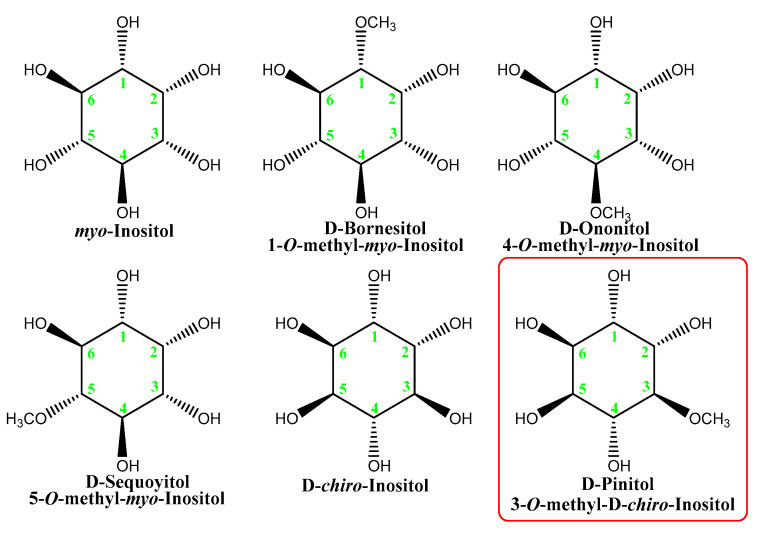
Inositols and their methyl ethers isolated from Carob.

**Figure 8 nutrients-14-01453-f008:**
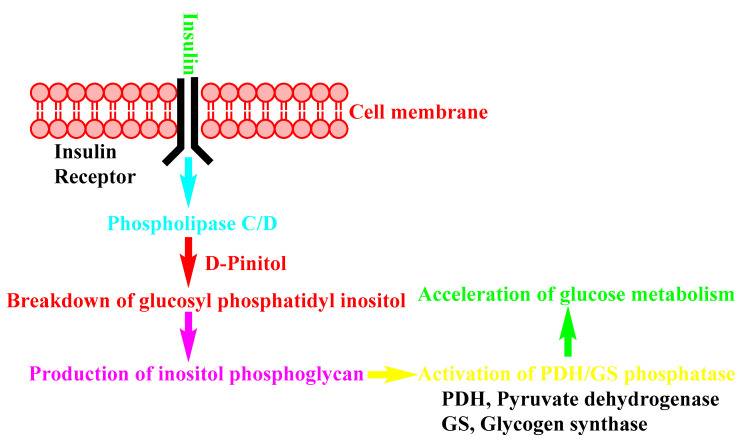
Insulin-sensitizing mechanism of D-Pinitol.

**Figure 9 nutrients-14-01453-f009:**
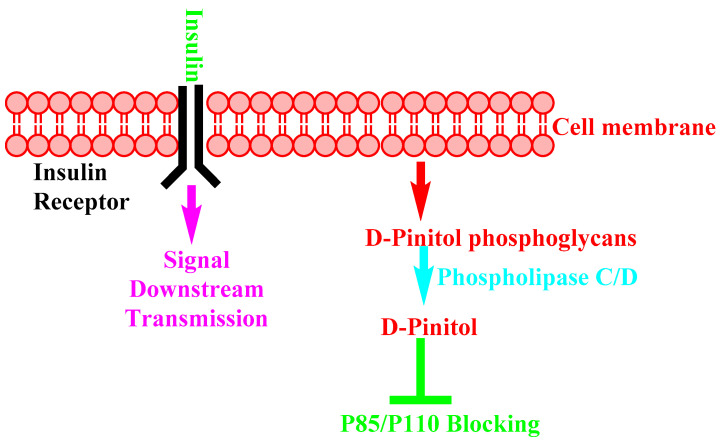
Insulin-mimetic mechanism of D-Pinitol.

**Table 1 nutrients-14-01453-t001:** General composition of Carob fruits [2].

Component	Proportion (%)
Moisture	6.3–7.6
Protein	1.7–5.9
Ash	2.3–3.2
Fat	0.2–4.4
Total dietary fiber	11.7–47
Starch	0.1
Total carbohydrates	42–86
Fructose	2–7.4
Glucose	3–7.3
Sucrose	15–34
D-Pinitol	5.5

**Table 2 nutrients-14-01453-t002:** Functions of insulin in human body [21].

Effect Type	Role of Insulin
Metabolic	Stimulation of glucose transport and metabolism
	Stimulation of glycogen synthesis
	Stimulation of lipogenesis
	Inhibition of lipolysis
	Stimulation of ion flux
Growth-promoting	Stimulation of DNA synthesis
	Stimulation of cell growth and differentiation
Metabolic & Growth-promoting	Stimulation of amino acid influx
	Stimulation of protein synthesis
	Inhibition of protein degradation
	Stimulation of RNA synthesis

**Table 3 nutrients-14-01453-t003:** Molecular mechanisms of insulin resistance [24].

Molecular Mechanism	Roles in Insulin Resistance
Upregulation of PTP1B [25]	Reverses insulin-induced phosphorylation in tyrosine residues of IRS-1 and so impairs insulin signal transduction
Inflammatory mediators and adipokines	Activation of IKKβ/NF-κB and JNK pathways, serine phosphorylation of IRS-1 in the site of 307, declines GLUT-4 expression, reduces IRS-1 expression via ERK1/2, induce IRS degradation through SOCS1- and SOCS3-dependent mechanisms
Free radical overload	Activates several serine–threonine kinase pathways, i.e., IKKβ/NF-κB and JNK, IRS degradation, suppresses GLUT-4 expression and localization in cell membrane, decreases insulin-induced IRS-1 and PIP-kinase relocation between cytoplasm and microsomes, decreases PKB phosphorylation, serine phosphorylation at site of serine 307 of IRS-1, activates inflammatory responses
Defects in serine phosphorylation of IRS-1	Decrease in insulin receptor phosphorylation, phosphorylation in serine 307 which blocks signaling
Obesity and adipocytes importance	Decrease in insulin receptor phosphorylation, phosphorylation in serine 307 which blocks signaling
Accelerated insulin degradation	Autoimmune antibodies against insulin or abnormal insulin structure due to mutation
Mitochondrial dysfunction	Induces oxidative stress, impairs insulin signaling
Reduced the capacity of receptors to binding to insulin	Decrease in number of insulin receptors, reduction in functional receptors due to mutation, autoimmune antibodies against insulin receptors
Mutations of GLUT-4	Point mutation changes normal modification of GLUT-4, inhibits glucose entering into dependent cells and impairs subsequent signaling pathways
ER stress	Disrupts proper protein folding leading to accumulation of misfolded proteins

PTP1B [25], protein tyrosine phosphatase 1B; IRS-1, insulin receptor substrates-1; IKKβ/NF-κB, central regulator of NF-κB; GLUT-4, type 4 glucose transporter; ERK, extracellular signal-regulated kinase SOCS1/3, suppressor of cytokine signaling; JNK, c-Jun N-terminal kinase; ER, endoplasmic reticulum.

**Table 4 nutrients-14-01453-t004:** Selected publications of insulin regulation and women fertility disorders treatment of inositols.

Property Short Description	Type of Publication	Ref., Year
Insulin regulation in human diabetics	research	[39], 1990
Treatment respiratory disorders in infants	research	[40], 1992
Insulin regulation in human diabetics	research	[41], 1993
Treatments of psychiatric disorders	review	[42], 1997
Treatment of polycystic ovary syndrome (PCOS)	research	[43], 1999
Treatment of Alzheimer disease, in vitro	research	[44], 2000
Insulin regulation in human diabetics	research	[45], 2005
Treatment of endothelial dysfunction, antioxidant, animal model	research	[46], 2006
Biological roles	review	[47], 2007
Derivatives and their functions	review	[48], 2008
Treatment of PCOS	review	[49], 2014
Insulin regulation in obese male children	research	[50], 2016
Treatment of PCOS	review	[51], 2016
Treatment of PCOS	research	[52], 2017
Bioavailability for treatment of PCOS	review	[53], 2017
Treatment of PCOS in subfertile women	review	[54], 2018
Effects on glucose homeostasis	review	[55], 2019
General presentation of medicinal activities	review	[56], 2019
Treatment of PCOS	review	[57], 2020
Treatment of PCOS, with other technologies	review	[58], 2021
Treatment of preterm birth	review	[59], 2021
Treatment of psychological symptoms in PCOS	review	[60], 2021
Insulin regulation in pregnancy	review	[38], 2022

**Table 5 nutrients-14-01453-t005:** Published properties of D-Pinitol.

Activity/Property	Testing Method	Ref.
Anti-Alzheimer	In vivo, mice	[67]
Anti-Alzheimer	In vitro, hippocampal cultures	[68]
Anti-Alzheimer	In vivo, *C. elegans*, mice	[69]
Antiaging	In vivo, *D. Melanogaster*	[70]
Antibacterial	*M. smegmatis*	[71]
Anticancer	In vitro, human cancer cells	[72,73,74,75,76,77]
Anticancer	In vivo, rats	[78,79,80,81,82,83]
Anti-colitis	In vivo, rats	[84]
Antidepressant	In vivo, mice	[85]
Antidiabetic	In vivo, mice/rats	[86,87,88,89,90,91,92]
Antidiabetic	In vivo, humans	[93,94,95,96,97,98]
Antidiabetic	Theoretical evaluation	[99]
Antidiarrheal	In vivo, mice	[100]
Antifibrotic	In vivo, mice	[101]
Antihyperlipidemic	In vivo, rats	[64,102]
Anti-inflammatory	In vivo, mice/rats	[103,104,105,106]
Anti-inflammatory	In vitro, Human cells	[72,107,108]
Anti-inflammatory	In vitro, BV2 microglial cells	[109]
Antinociceptive	In vivo, mice	[100]
Anti-obesity	In vivo, humans	[110]
Anti-obesity	In vivo, rats	[111]
Anti-osteoclastic	In vitro, UAMS32 cells	[112]
Antioxidant	In vivo, rats	[78,81,82,88,113]
Anti-psoriatic	In vivo, mice	[114]
Antiviral	Theoretical evaluation	[115]
Asthma treatment	In vivo, mice	[116]
Bone protection	In vitro, Bone marrow cell lines, rats	[117]
Bone protection	In vivo, rats	[118]
Cardioprotective	In vivo, humans	[93]
Cardioprotective	In vivo, mice/rats	[119,120]
Cytotoxic	In vitro, human cancer cell lines	[121]
Diuretic	In vivo, mice	[122]
Geno-protective	In vitro, monkey liver cell lines	[123]
Hepatoprotective	In vivo, humans	[124]
Hepatoprotective	In vivo, mice/rats	[125,126,127,128,129,130,131]
Hydration biomarker	In vivo, humans	[132,133]
Hypotensive	In vivo, mice	[134]
Immuno-protective	Theoretical evaluation	[99]
Immuno-protective	In vivo, mice	[116,135,136]
Immunosuppressive	In vivo, mice	[137]
Insulin regulation	In vivo, mice/rats	[111,131,138,139,140,141]
Insulin regulation	In vivo, humans	[96,142]
Insulin regulation	In vitro, 3T3-L1, HUVEC cells	[143,144]
Memory enhancement	In vivo, rats	[90]
Nanoparticles loaded	In vitro, against *M. smegmatis*	[29]
Nephroprotective	In vivo, mice/rats	[105,145]
Neuroprotective	In vivo, mice/rats	[85,122,146,147,148]
Sleep enhancer	In vivo, *D. melanogaster*, in vitro PC12 cells	[149]
Synergism w/ curcumin	In vitro, PC12 cells, against As^+3^ toxicity	[150]
Wound healing	In vivo, rats, in vitro, HaCaT cells	[151]

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
