# Peer review of "D-Pinitol—Active Natural Product from Carob with Notable Insulin Regulation"

_nutrients, 2022, doi:10.3390/nu14071453_

Round 1

Reviewer 1 Report

Although the topic is interesting, this manuscript should be reorganized. Not all sections are focused on D-Pinitol, such as Section 4 and 5. They should be introduced as background, not as major contents.

Other concerns:

(1). The references follow the titles of most tables and figures. Is this means the whole table or figure were cited, not summarized by the authors?

(2) The title may be incorrect.

Reviewer 2 Report

The publication written by Abdullatif Azab presents D-pinitol and its properties in the context of obtaining this compound from carob - plant, which can be one of D-pinitol the most important sources. The topic seems to be interesting and, at the same time, rarely discussed among authors. 

It is worth noting that the paper contains many new literature items from the last 5 years. It shows that the author is highly prepared for the topic, which is a big plus. Moreover, in my assessment, the publication should be qualify rather as a minireview, because it actually summarizes the latest news in a short and understandable way. 

However, in my opinion, if the publiction  is going to be published, it should be redrafted. 

First, whole paper is rather based on other review publications, while research items should be used in the most part of publication. In particular, when it comes to the properties of inositols, presented in Table 2 - only research papers should be included.

Second, diagrams and tables take up too much space compared to the information available in the literature and to author's own comments. For example, Table 3 would be much more informative with comments and descriptions of the presented research papers. Then the reader would know more about the original studies after reading this publication. This is all the more justified as D-pinitol is the subject of this publication. 

Moreover, more than half of the figures and tables included in the publication come directly from other publications, without making any changes. I do not understand the sense of duplication the same schemas, after all, the reader may see the original paper, especially since the schema is not accompanied by its description or even an explanation of the used abbreviations. Wouldn't it be more professional to propose a diagram containing similar data, but developed and made by yourself?

Overall, the publication has potential and is interesting in terms of the source of D-pinitol. However, the author should make two major changes, i.e.:
- developing paragraphs related to the subject of the publication, e.g. using research papers cited in tables;
- replacing borrowed schemas with your own or resigning from some of them (maybe creating your own on the basis of several presented ones will a be better idea).

Reviewer 3 Report

The manuscript entitled "D-Pinitol – Active Insulin Regulator Inositol from Carob" summarizes the bioactivities already reported for D-pinitol, giving emphasis to its role as an insulin regulator. However, It is not clear if the author want to emphasize carob or D-pinitol. If the author's purpose is to highlight the importance of D-pinitol, I think that a table with different plants sources of D-pinitol is missing. I also think the author could have prepared his own figures for the paper, based on the references he cited.

Other minor suggestions are given below:

Line 40 - What do the authors mean by "new" compounds?

Figure 2 - Why is the figure of D-chiro-inositol with a different color, including the name of the compound?

Table 2 - Please, state the meaning of ED.

Line 72 - Please, replace D-Ssequoyitol by D-Sequoyitol.

Line 133 - The sentence "This importance emerges from three major" is incomplete.

section 5 - This section should be reduced or deleted because it is not related with carob compounds.

Discussion section - Please, add the botanical authority to all Latin names.

For all the reasons presented above, I recommend major revisions.

Round 2

Reviewer 2 Report

The changes proposed by the reviewers positively influenced the understanding and clarity of the manuscript. The author made visible changes to the article, and rejected some of them, arguing accordingly.

However, in my opinion, no modifications were made to the presented diagrams as the author claims. I understand that the borrowed diagrams will remain unchanged. Therefore, I recommend that the author adds a paragraph under each of them with his own interpretation of the scheme. 

Author Response

28/03/2022

Dear Reviewers,

ALL figures in the article are original.

Explanations were added to/in figures 2,8,9 that were modified according to your comments.

Sincerely,

Dr. Abdullatif Azab

Reviewer 3 Report

All figures should be original and not taken from reference s

Author Response

(The authors gave the same response as above.)
